Calcium affinity of human α-actinin 1

Backman Lars lars.backman@chem.umu.se
Department of Chemistry, Umeå University , Umeå , Sweden
Uversky Vladimir
Electronic publication date: 2015 May 7
Publication date: 2015
Volume: 3
Electronic Location ID: e944
Received 2015 Mar 11; Accepted 2015 Apr 17
Copyright: © 2015 Backman
Copyright year: 2015
Copyright holder: Backman
License: This is an open access article distributed under the terms of the Creative Commons Attribution License, which permits unrestricted use, distribution, reproduction and adaptation in any medium and for any purpose provided that it is properly attributed. For attribution, the original author(s), title, publication source (PeerJ) and either DOI or URL of the article must be cited.
License URL: https://creativecommons.org/licenses/by/4.0/

Keywords: EF-hand, α-actinin, calcium binding

Funding: Carl Tryggers Stiftelse This work was supported by grants from Carl Tryggers Stiftelse. The funders had no role in study design, data collection and analysis, decision to publish, or preparation of the manuscript.

==============================
Due to alternative splicing, the human ACTN1 gene codes for three different transcripts of α-actinin; one isoform that is expressed only in the brain and two with a more general expression pattern. The sequence difference is located to the C-terminal domains and the EF-hand motifs. Therefore, any functional or structural distinction should involve this part of the protein. To investigate this further, the calcium affinities of these three isoforms of α-actinin 1 have been determined by isothermal calorimetry.

Introduction

Calcium ions are versatile secondary messengers influencing numerous cellular processes in eukaryotic organisms. A common target is calmodulin that upon binding of calcium ions undergoes a conformational change that allows it to interact with and regulate a large number of proteins, notably many enzymes involved in various metabolic pathways (Wang & Waisman, 1979; Klee, Crouch & Richman, 1980; Chin & Means, 2000; Kortvely & Gulya, 2004; Schaub & Heizmann, 2008). Another target for calcium is α-actinin that is involved in regulation and control of the intracellular infrastructure through its ability to interact with actin filaments (Sjöblom, Salmazo & Djinovic-Carugo, 2008). Similar to calmodulin and many other calcium-binding proteins, an EF-hand motif constitutes the binding site for calcium ions also in α-actinin.

All vertebrates have four genes coding for α-actinin, except for birds that appear to have only three (Virel & Backman, 2007). Two genes (ACTN2 and ACTN3) give rise to the calcium-independent isoforms (α-actinin 2 and α-actinin 3) that generally are found in muscle cells. The other two genes (ACTN1 and ACTN4) code for calcium sensitive isoforms (α-actinin 1 and α-actinin 4) that are present in non-muscle cells (Burridge & Feramisco, 1981; Noegel, Witke & Schleicher, 1987; Blanchard, Ohanian & Critchley, 1989; Foley & Young, 2014). Invertebrates, fungi and protozoa express in most cases a single isoform of α-actinin. Plants as well as algae and some fungi appear to lack a α-actinin gene (Virel & Backman, 2004).

All α-actinins have a common structure: a N-terminal actin-binding domain, comprising two calponin-homology domains, a rod domain comprising one to four spectrin repeats and at the C-terminal an EF-hand domain, containing 4 plausible EF-hand motifs, that in some isoforms bind calcium ions (Blanchard, Ohanian & Critchley, 1989; Otey & Carpen, 2004; Sjöblom, Salmazo & Djinovic-Carugo, 2008). Due to the ability to form antiparallel dimers, α-actinin is able to cross-link actin filaments into bundles or loose networks as well as to attach filaments to membranes and other structures (Wachsstock, Schwarz & Pollard, 1993; Luther, 2000; Foley & Young, 2014). In the antiparallel dimer, the EF-hand domain in one monomer is placed close to the calponin-homology domains in the other monomer. It has been shown that calcium reduces the affinity for actin filaments (Rosenberg, Stracher & Burridge, 1981; Virel & Backman, 2006), therefore it is possible that binding of calcium induces a conformation change in the EF-hand motif that in turn affects the actin binding.

The non-muscle α-actinins are intimately connected to cell motility through their roles in lamellipodia formation and focal adhesion as well as force generation. Therefore, it is not surprising that the expression levels of either α-actinin 1 or 4 have been correlated to tumor cell proliferation and invasiveness. However, the understanding is far from complete as the correlation appears to depend on the cell type. For instance, downregulation of α-actinin 1 does not affect motility of astrocytoma cells (Quick & Skalli, 2010) but reduces that of T lymphocytes (Stanley et al., 2008). Similar somewhat contradictory results have been observed also for α-actinin 4 (Hara et al., 2007; Welsch et al., 2009; Quick & Skalli, 2010).

The expression pattern of α-actinin 1 is rather complex due to alternative splicing of two exons (19a and 19b) in the EF-hand domain (Waites et al., 1992; Imamura et al., 1994). Inclusion of exon 19a but not 19b in the transcript leads to expression of a non-muscle α-actinin 1 with an active EF-hand domain. Expression of a transcript including exon 19b but not 19a gives rise to a calcium-insensitive smooth muscle isoform. In addition to these two transcripts that are expressed in most tissues, a brain specific transcript has been identified (Kremerskothen et al., 2002). This transcript includes both exon 19a and 19b and is expressed only in brain and, in particular, in hippocampus. Thus, apart from the first EF-hand motif, the sequences of the three isoforms are identical.

Interestingly, exon array analysis of colon, bladder and prostate cancer indicates that the α-actinin 1 gene ACTN1 is spliced differently in normal and tumor cells. In normal cells the smooth muscle isoform dominates whereas in cancer cells the smooth muscle isoform is downregulated and the non-muscle isoform dominates (Gardina et al., 2006; Thorsen et al., 2008).

Since the only sequence difference between these three isoforms is in the first EF-hand motif, any functional diversity must be related to this part of the protein. It is likely that the splicing of exon 19a in the smooth muscle isoform leads to a calcium-insensitive variant as this removes part of the first EF-hand motif. It is also possible that the sequence differences lead to different folding of the domain. Therefore, to investigate possible functional and structural differences, the EF-hand motif of these α-actinin isoforms were cloned, expressed and characterised by spectroscopy and isothermal calorimetry.

Materials and Methods

Cloning, expression and purification

The gene fragment coding for residues 712–914 (base pair 2444–3055) of human α-actinin 1 (accession NM_001130004) was obtained from GenScript, USA. This fragment contains exons 18–21 of human α-actinin 1 and codes for the brain isoform. The non-muscle isoform, lacking exon 19b, and the smooth muscle isoform, lacking exon 19a, were obtained by mutagenesis of the original gene fragment (Fig. 1). Mutagenesis and control of fidelity of gene fragments were performed by GenScript, USA.

Figure 1 Exon organisation of the C-terminal domain.

Alternative splicing of exons 19a and b, give rise to three different transcripts of human α-actinin 1. The non-muscle isoform lacks exon 19b whereas smooth muscle α-actinin 1 lacks exon 19a. The brain isoform contains both exon 19a and 19b.

The gene fragments were excised from pUC57 by BamHI and XhoI digestion and ligated into pET-TEV (a modified pET-19b) vector containing an N-terminal 10xHis-tag and a TEV protease cleavage site. This produced three plasmids: pET-BR-ACTN1-EF (brain), pET-NM-ACTN1-EF (non-muscle) and pET-SM-ACTN1-EF (smooth muscle).

E. coli BL21(DE3) cells were transformed (by heat-shock) with the purified plasmids containing the different constructs. The transformed cells were cultured at 37 °C in Luria-Betani medium containing 100 µM carbenicillin until an optical density of 0.6–0.8 at 600 nm was reached. Protein expression was induced by addition of isopropyl thio-β-D-galactoside to a final concentration of 0.5 mM and cells were grown overnight at 16 °C. Cells were harvested by centrifugation (50,000 × g for 30 min), resuspended in 25 mM sodium phosphate buffer, pH 7.6, 150 mM NaCl and stored at −20 °C. Frozen cells were thawed and then lysed by sonication. Triton X-100 and polyethyleneimine were added to final concentrations of 1% and 0.05%, respectively. After 30 min on ice, cell debris was removed by centrifugation at 50,000 × g for 30 min. Imidazole was added to the clarified supernatant to give 10 mM final concentration and loaded onto a HiTrap™ Chelating HP column (GE Healthcare Life Sciences, Little Chalfont, UK) charged with nickel. Unbound proteins were eluted with 25 mM sodium phosphate buffer, pH 7.6, 150 mM NaCl, 10 mM imidazole. Bound proteins were eluted with an imidazole gradient ranging from 10 to 510 mM imidazole in the same buffer. Imidazole in eluted fractions was removed by gel filtration on a HiPrep 26/10 desalting column (GE Healthcare Life Sciences, Little Chalfont, UK). Collected fractions were incubated at 4 °C for 16 h in the presence of Tobacco Etch Virus (TEV) protease (kindly provided by Dr. David S. Waugh). The released 10 × His-tag and the 6 × His-tagged TEV protease were removed by affinity chromatography as before. Finally, purified proteins were transferred into 50 mM Tris–HCl, pH 7.6, 200 mM KCl by gel filtration on HiPrep desalting columns (GE Healthcare Life Sciences, Little Chalfont, UK) and concentrated by Amicon Ultra 3 K centrifugal filter devices (Millipore Corporation, Billerca, Massachusetts, USA).

Protein concentration of native protein was determined from the absorbance at 280 nm using the molar absorptivity, as calculated from the amino acid sequence (using ProtParam at the ExPASy proteomics server). Purity of the expressed peptides were routinely determined under denaturating conditions by SDS-polyacrylamide gel electrophoresis (Laemmli, 1970).

CD and fluorescence spectroscopy

Far-UV circular dichroism (CD) spectra were acquired on a Jasco-J-810 spectrophotometer using a quartz cuvette with 2 mm path length. Spectra were collected between 200 and 260 nm with a data interval of 0.025 nm. The mean residue molar ellipticity ([Θ]MRW) was determined from three accumulated spectra.

The web service MLRC (Guermeur et al., 1999) at Pôle Bioinformatique Lyonnaise was used to predict the secondary structure. The relation −Θ222+300039000 (Morrow et al., 2000) was used to estimate the α-helical content.

Intrinsic fluorescence was determined on a Jasco FP-6500 spectrofluorometer (JASCO, Easton, Maryland, USA). Samples were excited at 280 nm and emission spectra between 310 and 600 nm were collected.

Calcium binding

The calcium affinity was determined by isothermal calorimetry using a MicroCal iTC200 (GE Healthcare Life Sciences, Little Chalfont, UK). For this the measuring cell was loaded with 0.3–0.6 mM protein in 50 mM Tris–HCl, pH 7.6, 200 mM KCl, and the syringe was filled with 5–10 mM calcium chloride in the same buffer.

Residual calcium content in the concentrated protein samples was estimated using Quin2 (Linse et al., 1987). It was found to be less than 0.02 mol of Ca2+ per mol of protein, independent of whether the protein solutions had be decalcificated by passing them over a Chelex 100 column (Bio-Rad Laboratories, Hercules, California, USA). The calcium concentration in the measuring buffer was around 1 µM.

Injection profiles were analysed using the standard single-site model of Origin (MicroCal, GE Healthcare Life Sciences, Little Chalfont, UK) as well as of Sedphat (Houtman et al., 2007).

Fluorescence and CD spectroscopy were used to study the effect of calcium ions on the spectral properties of the peptides.

Results

Cloning, expression and purification

Transformation of E. coli with either of the plasmids, followed by induction and expression produced large amounts of protein. After purification on nickel-columns, cleavage by TEV protease and gel filtration all three proteins were purified to nearly homogeneity as determined by denaturing gel electrophoresis.

Initially, proteins were after the final gel filtration step kept in a Tris-buffered sodium chloride solution. Preliminary experiments by fluorescence spectroscopy indicated that the brain EF-hand peptide unfolded slowly with time at room temperature in this media. However, when the peptide was transferred into a Tris-buffered potassium chloride solution the folding appeared stable as the emission spectra did not change with time. Therefore, the final gel filtration step was used to transfer the purified proteins into this buffer.

CD and fluorescence measurements

To check whether the purified peptides were folded CD measurements in the far-UV range were used. Figure 2 shows that each of the three peptides displayed the typical negative peaks at 208 and 222 nm observed for proteins with high α-helical content. Estimating the helical content from the mean residue molar ellipticity at 222 nm indicated helical contents of 45.1% (brain), 52.9% (non-muscle) and 47.9% (smooth muscle). These values agreed reasonable well with the predicted ones for the brain (49.3%) and non-muscle (51.4%) forms whereas the prediction for the smooth muscle form deviated more (54.5%) from the estimated value. In neither case, addition of calcium did not affect the CD spectra, independent of the concentration.

Figure 2 Far-UV CD spectra.

The far-UV CD spectra of the C-terminal domain of human brain (gold), non-muscle (black) and smooth muscle (red) α-actinin 1 were collected in 50 mM Tris-HCl, pH 7.6 and 200 mM KCl. The mean residue molar ellipticity was determined from three accumulated scans between 200 and 260 nm at 4 °C.

The π-π∗ transition of the amide at 208 nm is sensitive to inter-helix coupling. In contrast, the n-π∗ transition at 222 nm is more or less unaffected by inter-helix coupling but instead responsive to α-helical content (Lau, Taneja & Hodges, 1984; Zhou, Kay & Hodges, 1992). Therefore, the ratio of ellipticity at 222 and 208 nm has been used as a measure for formation of coiled-coils. Values of Θ222/Θ208 around or greater than 1 has been taken to indicate the presence of coiled-coils whereas a value around 0.83 would indicate non-interacting single-stranded α-helices. As the peptides exhibit ratios ranging from 0.88–0.93, it suggested that the α-helices in the EF-hand motifs are non-interacting. This would be expected assuming that the peptides fold into the typical helix-loop-helix structure of EF-hand motifs, with rather limited contacts between the helices.

Fluorescence spectroscopy is useful for monitoring structural changes (Eftink, 1991). The emission from excited aromatic residues, particular tryptophans, is very sensitive to changes in the surroundings. Thus, upon a conformation change leading to more or less exposure of the excited residue, the intensity of emission and/or the wavelength of emitted light will change. However, independent of the amount of added calcium to any of the α-actinin peptides no change in the emitted light could be detected. This was not particularly unexpected, as the only tryptophan present as well as all tyrosines are located outside the plausible calcium-binding loop (Fig. 4). Although there are several phenyalanine residues in the calcium-binding loop that may experience a change in the surroundings, the quantum yield from these are probably too low to be detected in the presence of other aromatic residues.

Isothermal calorimetry

To analyse the plausible calcium affinity of the α-actinin EF-hand domains in more detail isothermal calorimetry was used. Figure 3 shows typical heat traces and binding isotherms obtained from measurements of the different peptides. The results clearly showed that the brain and non-muscle isoforms both bound calcium in contrast to the smooth muscle isoform. Although measured heat of injection of the brain EF-hand domain was around 10 times as large as that of the non-muscle isoform, the estimated affinity of both were similar. Analysis of the binding isotherms using a one-to-one model gave dissociation constants of 49 ± 9 µM (11 measurements) and 56 ± 18 µM (12 measurements) for the brain and non-muscle isoforms, respectively. In both cases the number of sites were estimated to 0.9 ± 0.1. Interestingly, the entaphy and entropy changes differed substantially; ΔH and TΔS of the brain peptide was −3.7 and 2.2 kcal/mol compared to −0.34 and 5.46 kcal/mol for the non-muscle peptide.

Figure 3 Representative isothermal calorimetry thermograms for calcium binding to α-actinin 1 C-terminal domains.

Heat traces were obtained by injecting calcium chloride to brain (A), non-muscle (B) or smooth muscle (C) isoform, using the indicated concentrations of protein and calcium. After integration, a single-site-binding isotherm was fitted to the data to obtain the enthalpy of binding ΔH, entropy ΔS and the dissociation constant Kd. Best fitting dissociation constants for the calcium binding to the brain or non-muscle isoforms were 44.8 and 54.7 µM, respectively, for the data displayed in the figure. The first point in ITC titration was not included in the fitting. Data shown are representative of at least 10 independent experiments. In all figures, red circles represent experimental data and the full drawn line the best fit.

The heat trace obtained for the smooth muscle peptide was nearly identical to that obtained when injecting buffer without calcium, indicating that this EF-hand domain does not bind calcium under these conditions.

Sequence analysis

Analysis by Superfamily (Wilson et al., 2009) predicted that there should be 4 EF-hand motifs in the C-terminal domain. However, the E-values were only significant for the first EF-hand motif of brain and non-muscle α-actinin 1. Likewise, Pfam (Finn et al., 2014) recognized only the first EF-hand motif in the brain and non-muscle sequences as a proper calcium-binding motif. Pfam also identified motifs 3 and 4 as calcium-insensitive EF-hand motifs.

In an EF-hand loop, the calcium ion is coordinated in a pentagonal bipyramidal arrangement comprising 12 residues. In the typical coordination sphere, six residues at positions 1, 3, 5, 7, 9 and 12, denoted X, Y, Z, −Y, −X and −Z, coordinate the calcium ion. Residues X, Y, Z and −Z coordinate the calcium ion through side chain oxygenes whereas the −Y coordinate is provide by a backbone oxygene and the −X coordinate is provide by a water molecule interacting with the residue at this position.

As Fig. 4 shows, manual alignment of the calcium-binding loop of the consensus sequence with the different isoforms indicated that only EF-hand motif 1 of brain and non-muscle α-actinin contained the proper residues to coordinate a calcium ion. At position Y in the smooth muscle isoform, there is a lysine residue instead of a residue with an oxygene-bearing side chain; according to the consensus pattern (Kretsinger et al., 1991; Grabarek, 2006; Gifford, Walsh & Vogel, 2007) there is usually an aspartate residue or less often asparagine or serin residue at this position. In all other motifs, one or more of the oxygene-bearing residues required for coordinating calcium are missing.

Figure 4 Plausible calcium binding loops in the C-terminal domain of α-actinin 1.

The sequences of the C-terminal domain of brain, non-muscle and smooth muscle a-actinin 1 were aligned with the consensus sequence manually. In the calcium binding loop residues at the positions labelled X, Y, Z, −Y, −X and −Z coordinate the calcium ion, usually by side chain oxygenes. The residue at −Y coordinates the calcium ion through the back bone carbonyl oxygene. E, glutamate; G, glycine; I, isoleucine, leucine or valine; n, hydrophobic residue; ∗, any residue.

Discussion

The results show that the EF-hand motifs present in human brain and non-muscle α-actinin 1 bind calcium ions, with affinities around 45–55 µM, in contrast to the smooth muscle form of human α-actinin 1. This implies that transcription of exon 19a is prerequisite for calcium affinity and that there is only a single site for calcium in the C-terminal domain of α-actinin. This conclusion is supported by sequences analysis. Only the first EF-hand motif of brain and non-muscle isoforms contains the residues required for coordinating the calcium ion. In the other EF-hand motifs, several of the proper coordinating residues are missing (Kretsinger et al., 1991; Grabarek, 2006; Gifford, Walsh & Vogel, 2007).

The C-terminal domain of all vertebrate α-actinin s is extremely well conserved. Between human and coelacanth (Latimeria chalumnae) α-actinin 1 there are only three of 181 residues that differ. Even if the difference between α-actinin 1 and α-actinin 4 is greater, the sequence identity is still around 90% or more. Therefore, it seems very likely that there is only a single site for calcium in any calcium-sensitive vertebrate α-actinin. In contrast, Dictyostelium α-actinin appears to have two sites for calcium, one site with high affinity and a low-affinity site required for actin cross-linking activity (Noegel, Witke & Schleicher, 1987; Witke et al., 1993).

Previously, the calcium affinity of α-actinin, in terms of binding constant and stoichiometry, has only been determined for chicken gizzard (Wenegieme, Babitch & Naren, 1994) and rabbit macrophage α-actinin (Bennett, Zaner & Stossel, 1984). In both cases, the determined dissociation constants were in the low micromolar range (6 and 4 µM, respectively) and the number of binding sites was estimated to around 4 per dimer. Since the chicken gizzard α-actinin lacks the proper calcium-coordinating residues in the EF-hand motif, the observed binding indicated the presence of other binding sites or alternatively, was artifactual. Later it has been suggested that the observed high-affinity binding could have been due to EGTA bound to the protein (Pacaud & Harricane, 1993). It should also be noted that all previous measurements were done using full-length α actinins, in contrast to this study. Therefore it cannot be excluded that formation of an antiparallel dimer somehow induces a conformational change that increases the calcium affinity.

Both α-actinin 1 and 4 are generally described as calcium-regulated cross-linkers of actin filaments although the amount of calcium required appears to depend on the α-actinin isoform. 1.6 µM calcium reduces the cross-linking activity of Dictyostelium α-actinin to about half (Condeelis & Vahey, 1982) whereas 0.1 mM calcium reduces that of rabbit macrophage and human platelet α-actinin strongly (Bennett, Zaner & Stossel, 1984; Landon et al., 1985). More recently, it has been shown that the interaction between actin filaments and human non-muscle α-actinin is reduced by around 50% in the presence of 10 µM calcium or higher (Foley & Young, 2013).

It has also been observed that the calcium affinity as well the effect of calcium on binding and cross-linking activity of α-actinin is dependent on both pH and ionic conditions. The effect of pH appears to depend on the isoform of α-actinin; maximum cross-linking activity of Dictyostelium α-actinin occurs at pH 6.8 but is reduced with increasing pH and is nearly diminished pH 8 (Condeelis & Vahey, 1982). In contrast, the cross-linking activity of rabbit non-muscle α-actinin is more or less constant between pH 6.8 and 7.5 (Pacaud & Harricane, 1993). It has also been noticed that physiological ionic conditions, i.e., high concentration of KCl reduces or even diminish cross-linking completely (Rosenberg, Stracher & Burridge, 1981; Pacaud & Harricane, 1993). It has also been observed that binding to the high-affinity site in chicken α-actinin was eliminated by 120 mM KCl and binding could only be detected to low-affinity sites, with a binding constant of 166 µM (Wenegieme, Babitch & Naren, 1994).

Considering the distinct functional roles α-actinin 1 and 4 appear to have in adhesion and motility, and therefore in metastasis (Hara et al., 2007; Welsch et al., 2009; Quick & Skalli, 2010), surprisingly little is known about the physico-chemical properties of these proteins.

Supplemental Information

Supplemental Information 1 ITC raw data

Raw data accumulated from measurements of the calcium-binding to the three different isoforms of the C-terminal domain of human α-actinin 1, using the MicroCal iTC200 (GE Healthcare Life Sciences). In order to analyse the data it is necessary to extract the data from each sheet and use the appropriate software, such as Origin for ITC or Sedphat (Houtman et al., 2007).

In the fitting process, the first point in the ITC titration was excluded as well as any point that obviously was erroneous.

Reference: (Houtman et al., 2007) Studying multisite binary and ternary protein interactions by global analysis of isothermal titration calorimetry data in SEDPHAT: application to adaptor protein complexes in cell signaling. Protein Science 16:30–42.

Click here for additional data file.

Additional Information and Declarations

Competing Interests

Author Contributions

The authors declare there are no competing interests.

Lars Backman conceived and designed the experiments, performed the experiments, analyzed the data, contributed reagents/materials/analysis tools, wrote the paper, prepared figures and/or tables, reviewed drafts of the paper.

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
