# Peer review of "Calcium affinity of human α-actinin 1"

_PeerJ, doi:10.7717/peerj.944_

## Round 0.1 · original submission · Major Revisions

Please address all critical points raised by both reviewers.

Reviewer 1 ·

Basic reporting

No comments

Experimental design

The authors should provide more detailed information on the calcium binding experiments. What method did they use to remove calcium from protein samples before ITC Ca-titration experiments? What was calcium content in the Ca-freed samples? How did they monitor Ca content of the samples? What was Ca content in the samples in the CD measurements? Did Ca binding change the protein secondary structure?

Validity of the findings

It's hard to say anything about validity of the finding without answers on the above questions...

Additional comments

Essentially more information should be provided concerning experimental details of the Ca-titration experiments.

Reviewer 2 ·

Basic reporting

The article is well written. There are a few grammatical and spelling errors, but nothing that hinders a reader from understanding the research.

Line 51: 'reduced' should read 'reduces.'
Line 109-110: incomplete grammar. '...from of three accumulated spectra.'
Line 138: 'Was' should read 'were.'
Line 217: The grammar is awkward here. Deleting 'Later' will make it read better.

Experimental design

The experimental design is sound.

Validity of the findings

When determining the protein concentration, was the protein denatured when making this measurement? The conditions should be stated.

On line 165 it is mentioned that 'Figure 3 shows typical heat traces and binding isotherms...' How many replicates were performed? This should be stated explicitly. It is curious that such a difference in enthalpies is observed for the brain and non-muscle isoforms of a-actinin1 although the binding affinities were essentially the same within the range of error. Given a fitting was performed that gave the K and deltaH (DH) value, the TdeltaS (TDS) value should be easily obtainable. This value should also be reported for both the brain and non-muscle isoforms. This will allow for a more thorough analysis of the data and may allow readers to further purse why two proteins which have identical binding loops and similar linkers (shown in Fig. 6) have such different enthalpies of binding.

In Figure 3a, 3b, and 3c, the number of points recorded in the heat traces (top) differs from that shown in the integrated peak areas (bottom) corresponding to the measured heat released for each injection. For instance, in Figure 3a there are 30 data points on the top trace and 37 on the bottom plot (in 3b 17 vs 18 and in 3c 20 vs 17). These should be the same unless data points were discarded. Those that were discarded (e.g. the first injection) should be mentioned. Also, I assume the red circles represent the raw data and the black lines the best fit to a one-to-one model - however, a guide to the symbols and colors used should be stated.

---

## Round 0.2 · accepted · Accept

I am thankful that you seriously considered all the critical points and revised the manuscript accordingly.